# Reliable Discrimination of Green Coffee Beans Species: A Comparison of UV-Vis-Based Determination of Caffeine and Chlorogenic Acid with Non-Targeted Near-Infrared Spectroscopy

**DOI:** 10.3390/foods9060788

**Published:** 2020-06-16

**Authors:** Adnan Adnan, Marcel Naumann, Daniel Mörlein, Elke Pawelzik

**Affiliations:** 1Division Quality of Plant Products, Department of Crop Sciences, University of Goettingen, Carl-Sprengel-Weg 1, 37075 Goettingen, Germany; adnan.adnan@agr.uni-goettingen.de (A.A.); epawelz@gwdg.de (E.P.); 2Department of Animal Sciences, University of Goettingen, Albrecht-Thaer-Weg 3, D-37075 Goettingen, Germany; daniel.moerlein@uni-goettingen.de

**Keywords:** Arabica, Robusta, caffeine, chlorogenic acid, linear discriminant analysis, food fraud

## Abstract

Species adulteration is a common problem in the coffee trade. Several attempts have been made to differentiate among species. However, finding an applicable methodology that would consider the various aspects of adulteration remains a challenge. This study investigated an ultraviolet–visible (UV-Vis) spectroscopy-based determination of caffeine and chlorogenic acid contents, as well as the applicability of non-targeted near-infrared (NIR) spectroscopy, to discriminate between green coffee beans of the *Coffea arabica* (Arabica) and *Coffea canephora* (Robusta) species from Java Island, Indonesia. The discrimination was conducted by measuring the caffeine and chlorogenic acid content in the beans using UV-Vis spectroscopy. The data related to both compounds was processed using linear discriminant analysis (LDA). Information about the diffuse reflectance (log 1/R) spectra of intact beans was determined by NIR spectroscopy and analyzed using multivariate analysis. UV-Vis spectroscopy attained an accuracy of 97% in comparison to NIR spectroscopy’s accuracy by selected wavelengths of LDA (95%). The study suggests that both methods are applicable to discriminate reliably among species.

## 1. Introduction

The adulteration of *Coffea arabica* (Arabica) and *Coffea canephora* (Robusta) is a common problem in the coffee trade [1]. It results, among others, from the price difference between the species. Arabica receive more than 50% higher market price compared with Robusta. From 1990 to 2017, the average annual price of green beans of Arabica (US $2.51 per kg) was higher than that of Robusta (US $1.63 per kg). Arabica takes up approximately 58% of the global production of coffee compared with Robusta’s 42% global share. This implies that the consumption of Arabica is more preferable than Robusta [2]. Consequently, trade fraud involving the substitution of Arabica with Robusta cannot be avoided. Such fraud includes the addition of low-cost materials like coffee beans from different geographical regions or species without stating that in the product label [3].

Arabica and Robusta differ in several aspects—for example, taxonomic classification, morphology, bean size and color, chemical compounds, and sensory evaluation [4,5,6]. For example, the mean liking scores in a consumer test based on aroma, flavor, and mouthfeel led to significantly higher values for Arabica (6.0) in comparison with Robusta (4.4) based on a nine-point category scale where “1” means “not at all vivid” and “9” means “very vivid” [7]. The results of this study support the view that many consumers prefer Arabica to Robusta. For these reasons, Robusta is considered an adulterant for Arabica.

The conventional procedure to discriminate among species is based on a visual inspection of the size, shape, and color of the beans [8]. The limitation of this approach is that the physical characteristics of the beans differ considerably between species and variety due to various genotypes and environmental factors [5]. Another common method for differentiating among species is sensory testing [7,9]. The disadvantages of this approach are that trained panels are not always available and they are expensive [10]. A third disadvantage is that certain varieties of Arabica have sensory properties similar to those of Robusta—in terms of mouthfeel and bitterness; this would distort the test results [11].

Caffeine and chlorogenic acid play important roles for flavor formation and health effects on humans [12]. Caffeine consumption has well-known effects on the stimulation of brain functions and improvement in mood and physical performance; it is also associated with the reduction of the development of chronic degenerative diseases. However, excessive caffeine consumption may expose the drinker to cardiovascular disease and blood pressure problems [13]. Chlorogenic acid is a major component of green coffee beans and an important dietary polyphenol with potential health benefits, including anti-inflammatory, antidiabetic, anti-obesity, and anticarcinogenic effects [14].

Chromatographic techniques (e.g., high-performance liquid chromatography (HPLC) and gas chromatography (GC)) are commonly applied successfully to discriminate between coffee beans species [1,15]. Chemical compounds, for example, trigonelline, tocopherol, caffeine, and chlorogenic acid, are used to differentiate between Arabica and Robusta [16]. Trigonelline levels and the sum of α-, β-, and γ-tocopherols vary depending on the species. About 0.8% of the dry weight (dw) is composed of trigonelline and 0.028% of the sum of tocopherols in Arabica, which is higher than in Robusta (0.7% dw and 0.011%, respectively) [15,16]. Conversely, the caffeine (1.3% dw) and chlorogenic acid (8.1% dw) content in Arabica are lower than in Robusta (2.3% dw and 9.9% dw, respectively) [16].

Despite their accuracy, these chromatographic techniques involve equipment-intensive and time-consuming processes. Out of the different chemical compounds present in coffee, caffeine and chlorogenic acid were selected as key compounds in the present study; as mentioned, the concentrations of these compounds tend to differ across green coffee beans species.

An alternative to chromatographic techniques is ultraviolet–visible (UV-Vis) spectroscopy, which provides simplified measurement procedures that are time- and cost-effective [17]. To date, UV-Vis spectroscopy has been applied to measure the caffeine and chlorogenic acid content of coffee beans [18,19,20,21], and it has also been used to discriminate species on ground and roasted coffee. However, no previous study has so far used these compounds measured by UV-Vis spectroscopy to discriminate among green coffee beans species. We, therefore, investigated this approach as an alternative method to discriminate between Arabica and Robusta to help prevent fraud within the global coffee beans trade.

Previous studies have reported the applicability of NIR spectroscopy to discriminate between species [22,23]. However, they rarely covered aspects such as altitude or genotype that can cause considerable differences within species and variety [24]. In this study, we have therefore evaluated the capability of NIR spectroscopy to discriminate among the species of intact green beans from a different origin, variety, and altitude, in comparison with the UV-Vis-based determination of caffeine and chlorogenic acid. These methods allow high throughput and low involvement of labor—faster examination at a lower cost to discriminate species is preferable. Thus, the application of these methods may help prevent fraud in a desirable manner for the coffee industry, as compared with other existing methods.

## 2. Materials and Methods

### 2.1. Samples

Seventy-four green coffee beans samples from various locations on Java Island, Indonesia, were used in this study (Appendix A). The samples set of green beans represented different environmental factors, agricultural practices, and genetic characteristics, and were sourced from 38 processing facilities on Java Island during the harvesting season from July to August 2014. Of these 74 samples, 32 samples belonged to Arabica and 42 to Robusta. To ensure the authenticity of the samples, the coffee species were validated by agricultural extension officers and farmers.

The first step to obtain green beans was the harvest of the red coffee cherries from the coffee plantation surrounding the processing facility. After harvesting, the red cherries were processed (e.g., pulping, washing, drying) into green beans and stored in 60 kg bags. From these bags, the samples under study were collected randomly at 250 g per sample and were then transported using double-sealed plastic bags for analysis.

### 2.2. Determination of Caffeine and Chlorogenic Acid Content by UV-Vis Spectroscopy

The samples (*n* = 74) were prepared in line with the procedures used by Belay et al. and Navarra et al. [18,21]. First, the beans were freeze-dried (Epsilon 2-40, Christ, Hagen, Germany) and ground into a powder using a ball mill (Schwingmühle MM 400, Retsch, Haan, Germany). Next, the powder was screened through a 0.355 mm sieve. A total of 10 mg of sieved coffee powder was dissolved in 10 mL of distilled water. The solutions were stirred (550 rpm, 35 °C, 1 h) using a stirrer (Eppendorf ThermoMixer^®^ comfort, Eppendorf, Hamburg, Germany) and then passed through a paper filter (MN 615 1/4, Macherey-Nagel, Duren, Germany).

Caffeine extraction from the sample solution was performed by mixing 5 mL of filtrate with 5 mL of dichloromethane and stirring the liquid for one minute using a vortex mixer. Finally, the absorbance of caffeine in dichloromethane was measured by the UV-Vis spectrophotometer (HP 8453, Hewlett Packard, Boeblingen, Germany) within the range of 200–500 nm against the corresponding blank reagent (dichloromethane) and—as per to the Beer‒Lambert law—at a maximum wavelength of λ = 276 nm (Appendix A). The standard solutions were prepared by dissolving caffeine (anhydrous ≥98.5%, Carl Roth, Karlsruhe, Germany) in dichloromethane (Rotipuran ≥99.5%, Carl Roth, Karlsruhe, Germany) ranging from 10 to 35 ppm; the calibration yielded an R^2^ of 0.9974. The caffeine content was calculated in percent on a dry weight (dw) basis (Appendix A). Each measurement was replicated twice. The average standard deviation of the replicated caffeine determination was 0.1% dw.

The chlorogenic acid content was determined using a method similar to the one outlined for caffeine measurement, excluding the extraction step using dichloromethane [19,21]. The samples (*n* = 74) were prepared as follows: First, 1 mg of sieved coffee was dissolved in 10 mL of distilled water. The solutions were stirred for one minute and passed through a paper filter (MN 615 1/4, Macherey-Nagel, Duren, Germany). The chlorogenic acid was measured using UV-Vis spectroscopy against the corresponding blank reagent (distillate water), according to the Beer‒Lambert law at a maximum wavelength of λ = 324 nm (Appendix A). Dissolved chlorogenic acid (≥97% Carl Roth, Karlsruhe, Germany) in distilled water was used to prepare standard solutions in the range of 20 to 150 ppm (R^2^ of the calibration curve = 0.9999). The chlorogenic acid content was calculated in percent of dw based on two replications; the average standard deviation of duplicate measures was 0.9% dw (Appendix A).

### 2.3. Determination of Species by Near-Infrared Spectroscopy

The samples were analyzed using the NIR spectroscopy method, as previously reported [25]. A bench-top Fourier transform (FT-) NIR instrument with a rotating sample cup (Thermo Nicolet Antaris MDS, Thermo Fisher, Waltham, MA, USA) was used to acquire the diffuse reflectance spectra (log 1/R) of bulk samples of green coffee beans (40 g per sample) (*n* = 74) in a rotating sample cup (petri dish made of Schott Duran glass). The internal background spectra were collected once every hour. High-resolution diffuse reflectance (log 1/R) spectra at a wavelength range of 1000–2500 nm at 2 nm intervals were recorded as the averages of 64 scans. Each sample was replicated three times. Before conducting further calculations, the spectra were averaged.

### 2.4. Statistical Procedures

Statistical analysis was performed using R software (R Foundation, Vienna, Austria) for the UV-Vis spectroscopy method. The linear discriminant analysis (LDA) was carried out using Unscrambler^®^ X version 10.2 Network Client (CAMO software AS, Oslo, Norway). The box plot and Welch’s unequal variances *t*-test were used to explore the variability of caffeine and chlorogenic acid content among species [26,27]. The data of both compounds were then analyzed using LDA to discriminate among species. Cross-validation was performed to validate the results and stated as classification accuracy (in percent) [20]. The correlation between the altitude and the chemical compound was tested using Pearson’s product-moment correlation coefficient (*r*).

Multivariate analysis was carried out using the following procedure [25]. The first step in this analysis was to detect spectral outliers using principal component analysis (PCA) and Hotelling’s T^2^ ellipse 5% plot, based on the raw spectra of all samples (*n* = 74). Following the detection of spectral outliers, several preprocessing methods were applied (e.g., smoothing, the Savitsky–Golay derivative, normalization, baseline correction, orthogonal signal correction (OSC), multiplicative scatter correction (MSC), and extended multiplicative scatter correction (EMSC)). On the preprocessed spectral data, calibration (*n* = 49) and validation (*n* = 23) models were developed using partial least squares discriminant analysis (PLS-DA). Finally, all models were verified with regard to their prediction accuracy—that is, the number of latent variables (LVs), the coefficient of determination (*R^2^*), the root mean square error of calibration (RMSEC), and the root mean square error of prediction (RMSEP) (Appendix A) [28].

LDA was applied to the selected wavelengths of raw spectra. These selected wavelengths were derived from PLS-DA. The accuracy (in percent) was calculated using a full cross-validation procedure.

## 3. Results

### 3.1. UV-Vis Spectroscopy

#### 3.1.1. Caffeine Content

As shown in Figure 1a, the median and range values for caffeine content differed among species. The caffeine content in Arabica was significantly lower than that in Robusta (Welch’s unequal variances *t*(67.9) = −17.8, *p*-value < 0.001). On average, Arabica had a caffeine content of 1.8% dw, while Robusta contained 2.9% dw. The 95% confidence interval for the differences in the caffeine content among the species was between −1.3% dw and −1.0% dw.

#### 3.1.2. Chlorogenic Acid Content

The mean value of chlorogenic acid content in Arabica was 7.0% dw and in Robusta was 9.5% dw (Figure 1b). Arabica and Robusta differed significantly in their chlorogenic acid content according to Welch’s *t*-test (*t*(66.2) = −11.1, *p* < 0.001; 95% CI of mean difference: −2.0% dw and −1.9% dw), although few samples (5.4%) displayed similar levels of chlorogenic acid for both species.

#### 3.1.3. Discrimination among Species on the Basis of Caffeine and Chlorogenic Acid Content (by UV-Vis Spectroscopy)

As shown in the previous sub-section, discrimination among species on the basis of only a single chemical component (caffeine or chlorogenic acid) is unreliable. To discriminate among species, both compounds were used for a linear discriminant analysis (LDA). As shown in Figure 2, LDA discriminates between Arabica and Robusta with an accuracy of 97.3%. Only two samples of Arabica were mistakenly identified as Robusta samples. This shows that using both caffeine and chlorogenic acid content is a reliable method for discriminating among species.

### 3.2. Discrimination among Species Using NIR Spectroscopy

The following section examines the applicability of NIR spectroscopy. Raw diffuse reflectance (log 1/R) spectra (*n* = 74) were inspected to detect outlier data using principal component analysis (PCA) and projection of the Hotelling T^2^ ellipse (Figure 3a,b). Data points located outside the ellipse were considered to be spectral outliers and were removed. Two samples were spoiled microbially. Hence, they were deleted due to their potential negative influence on the model, though they may have contained valuable information regarding species variation [29].

Preprocessing methods were applied to raw spectra (*n* = 72) before using PLS-DA to discriminate among species. Table 1 shows that several PLS-DA models based on preprocessing methods produced fairly high *R^2^*. The PLS-DA models based on EMSC, normalization (area and mean), smoothing (moving average, three segments), and MSC yielded *R^2^* of the validation model of 81.3–90.5%; however, using various numbers of latent variables (LVs). Those models yielded higher accuracy compared with the validation model based on raw spectra of 71.5%. The details pertaining to the other PLS-DA models are presented in Appendix A.

Another significant aspect of the multivariate analysis is the ability to disclose wavelengths that are causative for discrimination and considered to be related to chemical compounds present in the sample. The weighted regression coefficient plot shows that several wavelengths (1212 nm, 1342 nm, 1465 nm, 1674 nm, 1929 nm, 2021 nm, and 2227 nm) contributed to the discrimination among coffee species (Figure 4a,b). These wavelengths are related to caffeine, chlorogenic acid, carbohydrates, sugars, trigonelline, lipids, water, proteins, and amino acids [30].

Subsequently, wavelengths identified to contribute to the PLS-DA discrimination among coffee species (Figure 4a,b) were used for LDA, which resulted in a classification accuracy of 95% (95% CI (87%, 98%)) (Figure 5). The LDA method using selected wavelengths performs with higher accuracy than the full spectra of PLS-DA. This result suggested that the selected wavelengths of 1212 nm, 1342 nm, 1465 nm, 1674 nm, 1929 nm, 2021 nm, and 2227 nm are satisfactory to discriminate between coffee species. When Arabica is considered the positive case to be detected, sensitivity and specificity of the test were 100% and 91%, respectively. Using MSC corrected spectra and the above seven wavelengths, the classification accuracy improved slightly to 97%.

## 4. Discussion

### 4.1. UV-Vis Spectroscopy

#### 4.1.1. Caffeine Content

A closer inspection of Figure 1a shows that one sample of Arabica exhibited caffeine levels similar to those in Robusta. These species have an overlapping range of 4.1% samples. Thus, merely relying on caffeine content to discriminate among species can lead to false classifications.

In accordance with the present results, a previous study has demonstrated that the caffeine content ranged from 0.8% to 1.8% dw (mean = 1.3% dw) in Arabica as compared with a range of 1.2% to 2.5% dw in Robusta (mean = 1.8% dw) [31]. This study also showed that the caffeine content values overlapped among species, which is consistent with other studies [32,33]. Taken together, this confirms that caffeine content is unsuitable as a single factor to discriminate between Arabica and Robusta.

A correlation test was conducted to obtain more insight whether the altitude as an environmental factor is responsible for the variability of caffeine content in the present study. The results showed a slightly negative but not significant correlation for Arabica of *r* = −0.25 (*p* = 0.17) and for Robusta of *r* = −0.11 (*p* = 0.50) (Appendix A). Other studies confirmed that environmental factors such as total irradiance, rainfall, temperatures, and potential evapotranspiration at various altitudes do not affect the caffeine content [34,35].

#### 4.1.2. Chlorogenic Acid Content

This result suggested that chlorogenic acid cannot be used as the only trait to discriminate among coffee species (Figure 1b). Accordingly, previous studies have demonstrated that the chlorogenic acid content values overlapped among species. In general, the chlorogenic acid content of green beans ranged between 4.0% dw and 8.4% dw in Arabica, and between 7.0% dw and 14.4% dw in Robusta [36].

Finding a correlation between chlorogenic acid and altitude is interesting because altitude may affect chlorogenic acid. A previous study showed lower chlorogenic acid on higher altitude [35]. The results of the present study showed that the chlorogenic acid content did not correlate with the altitude for Arabica (*r* = −0.25, *p* = 0.17) and Robusta (*r* < 0.01, *p* = 0.99) (Appendix A).

The multivariate statistical tools, that is, PCA and PLS-DA based on chlorogenic acid constituents (caffeoylquinic acid, feruloylquinic acid, and *p*-coumaroylquinic acid isomers), can be used for determining three coffee agricultural practices (organic, conventional, and biodynamic), but not for geographical identification of roasted coffee. The 5-caffeoylquinic acid and 4-caffeoylquinic acid were higher than 3-caffeoylquinic acid in all agricultural practices. Chlorogenic acid constituents are assumed to result from the absence of pesticide and pest-defense compounds [37].

#### 4.1.3. Discrimination among Species on the Basis of Caffeine and Chlorogenic Acid Content (by UV-Vis Spectroscopy)

Another advantage of measuring the caffeine and chlorogenic acid contents is that they are key compounds indicating the coffee flavor. Caffeine and chlorogenic acid are related to bitterness [38]. Thus, a higher level of these compounds may produce a more bitter coffee taste [30] and, subsequently, contribute to lower cup quality as determined by trained panelists [39]. This may explain why Arabica obtained a higher consumer liking than Robusta [7].

The present study indicates that UV-Vis spectroscopy can be used as an alternative approach to discriminate among species of green coffee beans on the basis of caffeine and chlorogenic acid content. The discrimination accuracy in our study is comparable to a previous study using caffeine, chlorogenic acid, trigonelline, total polyphenols, total free amino acids, and aqueous extract as determined by liquid chromatography (HPLC) for species discrimination [40]. A K-nearest neighbors’ classification resulted in an accuracy of 92.7–97.6%. The error rate occurred because some Arabica samples had an unusually high and some Robusta samples had an uncharacteristically low caffeine content [40]. As the present study uses less compounds, it can be considered comparable.

Recent research has shown that PLS-DA differentiated 100% of ground Arabica and blends. Caffeine, total soluble solids, quercetin-3-rutinoside, the Folin–Ciocalteu reducing capacity, and antioxidant capacity (DPPH assay) were the most discriminating variables. The classification of ground Arabica and blends was also obtained with 100% accuracy using LDA. Total flavonoids, Fe2+ chelating ability, quercetin-3-rutinoside, and total phenolic content were the analytical responses that discriminated between groups [41].

### 4.2. Discrimination among Species Using NIR Spectroscopy

The results show that the PLS-DA model based on MSC preprocessed spectra needed the lowest number of LVs. While the number of LVs was the lowest, the calibration and validation models based on MSC obtained neither the highest *R^2^* nor the lowest RMSEC and RMSEP compared with the other preprocessing methods. Simultaneously, the EMSC and normalization of the area and mean method yielded the highest *R^2^* (>90%) and the lowest RMSEC and RMSEP (<0.40), but the number of LVs is high (Table 1).

Selecting the ideal model is challenging because none of the models fit the criteria for the best model—that is, highest *R^2^*, lowest number of LVs, lowest RMSEC, and lowest RMSEP [42]. However, a low number of LVs is considered to produce a more robust model [43]. If a model is selected on the basis of different LV numbers, the *R^2^* value should be examined closely to avoid an overoptimistic model [44].

Thus, the PLS-DA model based on MSC processed spectra is selected here as the ideal model using full-range spectra for discriminating among green coffee beans species (Table 1). MSC (LVs = 3) reduces the number of LVs in the respective discrimination model compared with the model based on raw spectra data (LVs = 7). This indicates the presence of scattering and simultaneously shows that MSC can reduce the noise [11]. Scattering in NIR spectroscopy can be influenced by the sample traits and the measuring conditions [45]. Species variety can lead to different sizes and shapes [31,46]. As shown in Appendix A, the present study used green coffee beans samples of different species, and variety may lead to scattering problems. The beans were not screened to be of the same size—the aim was to create the actual sample conditions that prevail during trading.

However, using only selected wavenumbers and an LDA approach yielded a superior classification performance. This model was built using authentic sample conditions including samples with different sizes and shapes. The percent of correctly classified samples by the validation model was 95.5%. The classification performance based on selected wavelengths is more accurate than those of a previous study. Downey et al. [47] reported that the accuracy of NIR spectroscopy in discriminating among green coffee beans species ranges from 86.5% to 88.6%. The validation models were built using a factorial discriminant analysis with eight LVs, which is potentially an overfit model.

Recent study has demonstrated NIRS data using the typical band of the spectra, and the PLS-DA classifier predicts farming system (organic and conventional) of roasted coffee and provides results with 89% accuracy. The NIRS classification model, which is much simpler to develop and deploy, can provide good prediction with less instrumentation complexity and at a lower cost than proton transfer reaction mass spectrometry (PTR-MS) does. However, geographic identification was somewhat complex. The PTR-MS models using PLS-DA performed with slightly better accuracy than NIRS models (69% vs. 61%, respectively) [48].

According to the literature, wavelengths of 1209 nm, 1466 nm, 1726 nm, 1758 nm, 1904 nm, 2308 nm, and 2348 nm—relating to pure water and lipids—can be used to differentiate among green coffee beans species [47]. Another study suggested different selected wavelengths to discriminate among species of green beans. The wavelengths of 1671 nm, 1673 nm, and 2154 nm are associated with caffeine, and the wavelengths of 1778 nm, 1834 nm, and 2251 nm are associated with cellulose [22]. Two previously identified wavelengths (i.e., 1671 nm and 1673 nm) [16] are close to the wavelength selection of the experiment resulting model (1674 nm), which are associated with caffeine.

Buratti et al. [22] demonstrated that NIR spectroscopy offers 100% accuracy in discriminating between Arabica and Robusta. However, this study did not clarify the chemical composition of the beans. Hence, as shown in our study, accuracy may be compromised due to partly overlapping levels of caffeine and chlorogenic acid in both species.

Despite the potential of NIR spectroscopy, it is still a challenge to develop an applicable and sensitive method for discriminating among green coffee beans. As in the present study, the samples generally display variations among species and variety. Here, even samples from one particular location consist of several varieties because the farmers planted multiple batches at the same times. In addition to variety, models for species recognition could benefit from taking various environmental factors into account (Appendix A).

## 5. Conclusions

This study evaluated the applicability of UV-Vis spectroscopy and NIR spectroscopy to discriminate between the green coffee beans of Arabica and Robusta from Java Island, Indonesia. The results showed that both approaches are acceptable in terms of their classification accuracy. UV-Vis spectroscopy-based determination of two important compounds—that is, caffeine and chlorogenic acid—attained a slightly higher classification accuracy of 97.3%. NIR spectroscopy using seven selected wavelengths and LDA yielded a similarly high classification accuracy (95.5%). The findings suggest that, given both the speed, nondestructiveness and low involvement of labor of NIR spectroscopy, it is superior for on-site species discrimination. This study was limited by the environmental conditions and varieties of the beans samples. Thus, further research should include samples of different species and varieties from various coffee-producing locations worldwide in order to evaluate the robustness of NIR-based species discrimination.

## Figures and Tables

**Figure 1 foods-09-00788-f001:**
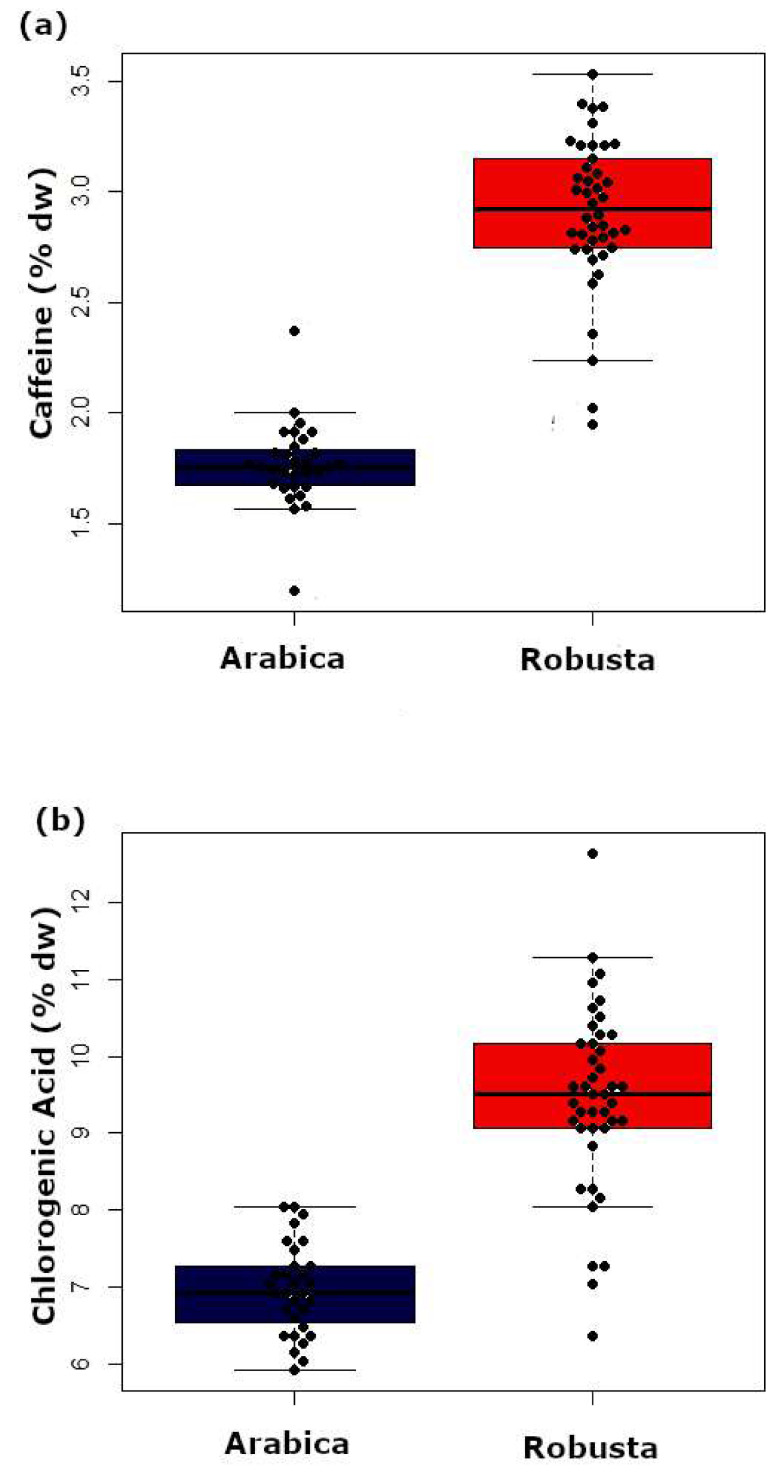
Caffeine content (**a**) and chlorogenic acid content (**b**) of green Arabica and Robusta beans obtained using UV-Vis spectroscopy (*n* = 74).

**Figure 2 foods-09-00788-f002:**
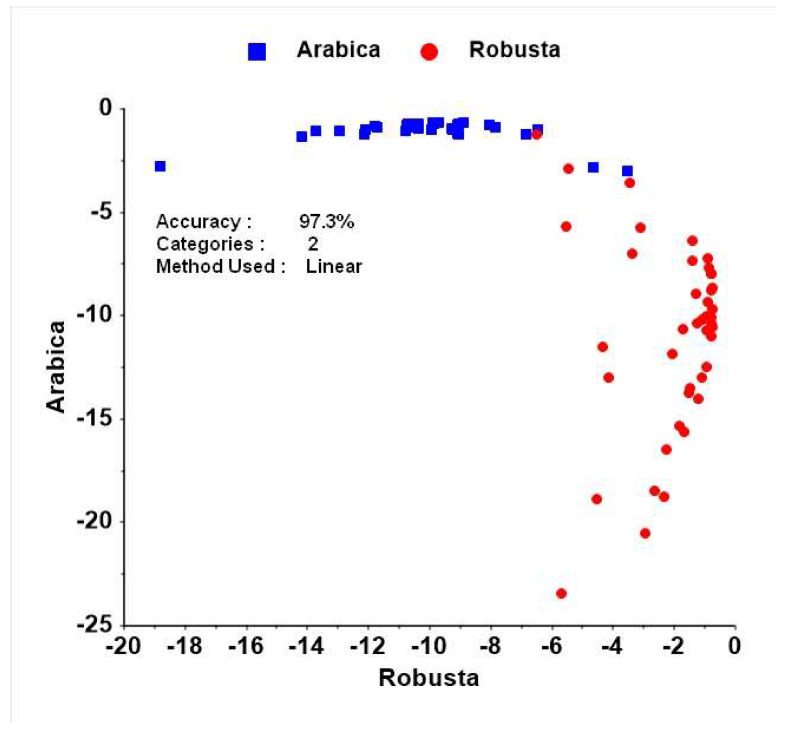
Discrimination among coffee species by linear discriminant analysis (LDA) using caffeine and chlorogenic acid derived from UV-Vis as predictor variables.

**Figure 3 foods-09-00788-f003:**
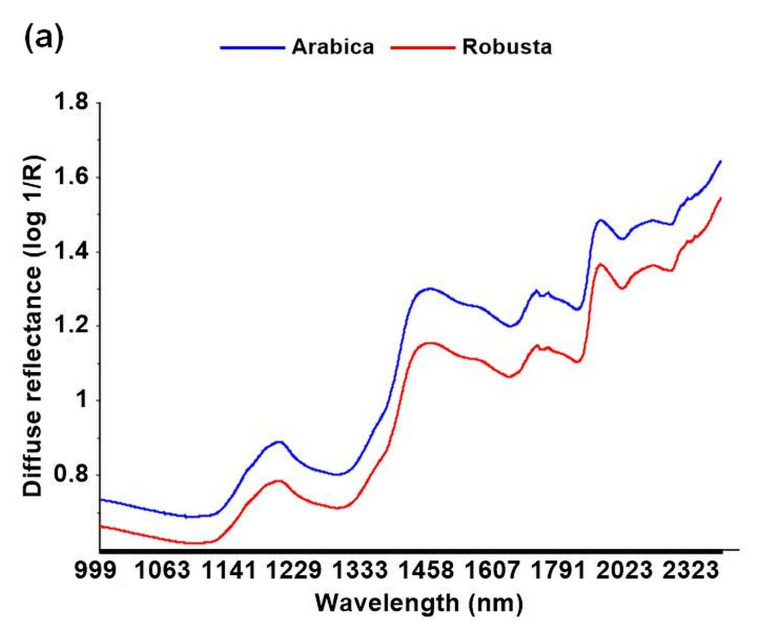
Mean diffuse reflectance (log 1/R) of raw spectra (**a**); score plot of principal component analysis using raw near-infrared spectra (log 1/R) with Hottelling’s T^2^ ellipse for outlier inspection. Samples outside the Hottelling’s T^2^ ellipse are considered spectral outliers (**b**).

**Figure 4 foods-09-00788-f004:**
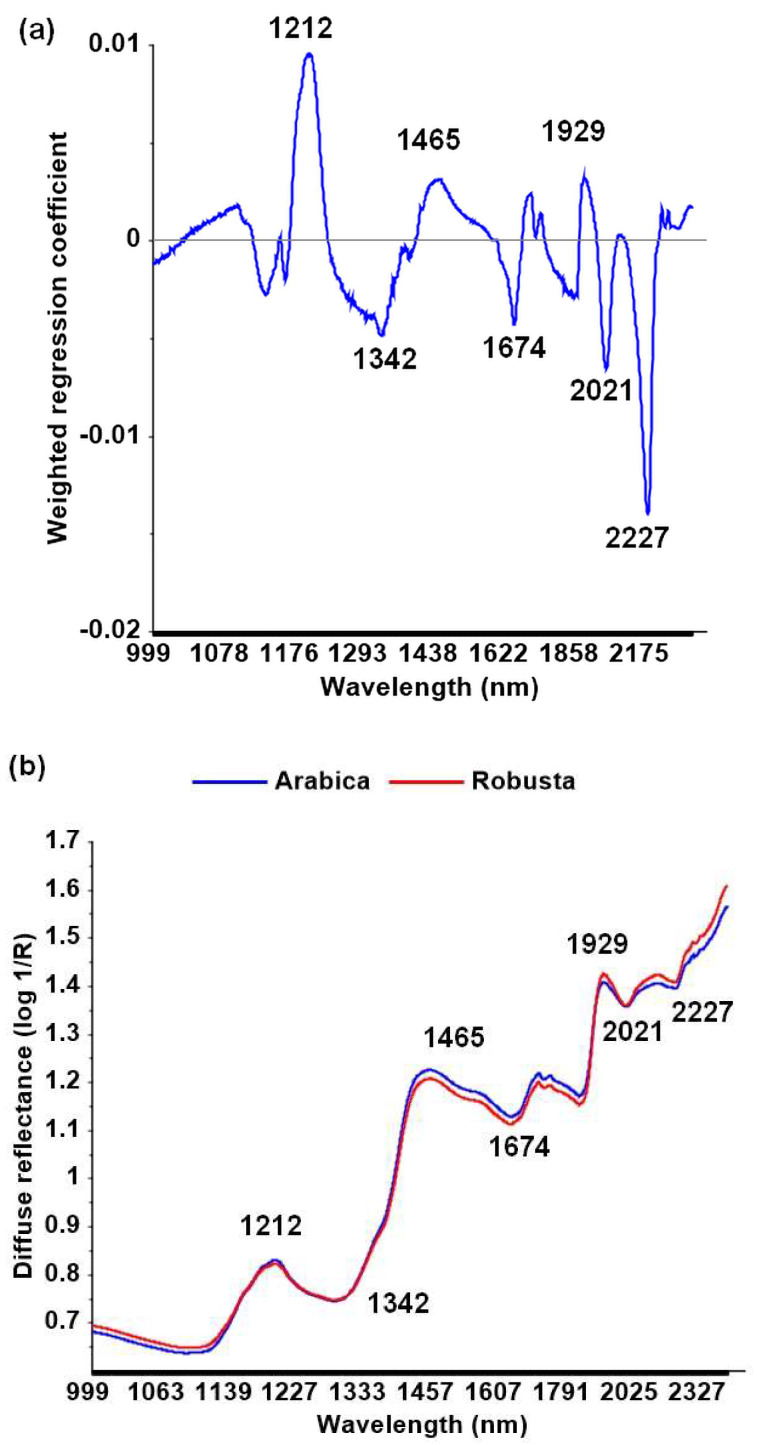
Weighted regression coefficient plot of the partial least squares discriminant analysis (PLS-DA) model based on multiplicative scatter correction (MSC) spectra of intact green beans (number of latent variables = 3) (**a**); mean diffuse reflectance (log 1/R) of MSC spectra (**b**).

**Figure 5 foods-09-00788-f005:**
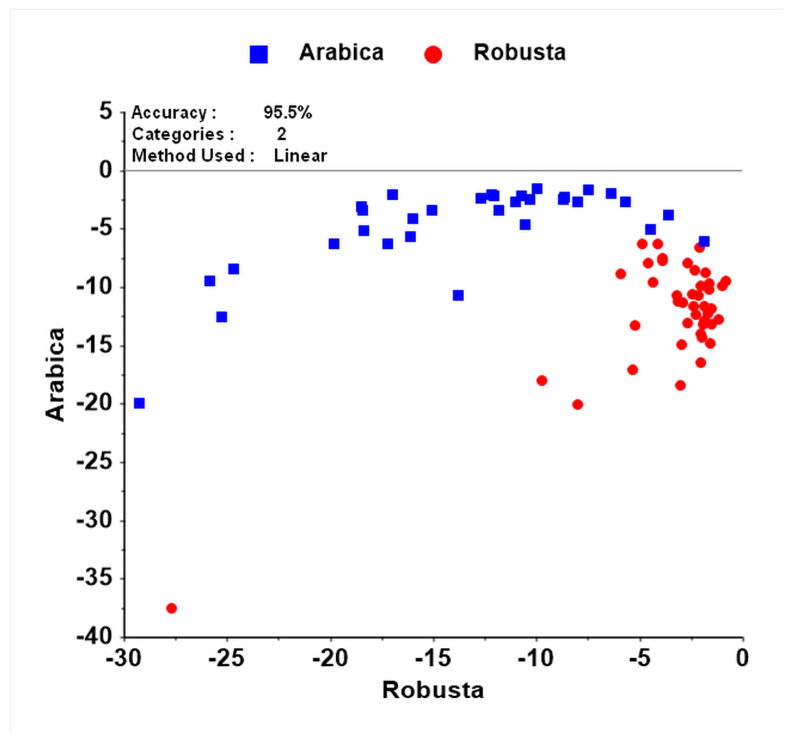
Discrimination among coffee species by linear discriminant analysis (LDA) using selected wavelengths derived from NIR spectroscopy as predictor variables (*n* = 74 samples).

**Table 1 foods-09-00788-t001:** Statistical parameters of partial least squares discriminant analysis models discriminate green coffee beans species using near-infrared spectra.

Preprocessing Method	LVs	R^2^ of Calibration Model (%)	RMSEC	R^2^ of Validation Model (%)	RMSEP
Raw	7	89.0	0.3266	71.5	0.6005
EMSC	6	91.4	0.2884	90.5	0.3641
Normalization (area)	7	93.2	0.2570	90.3	0.3745
Normalization (mean)	6	93.2	0.2570	90.3	0.3745
Smoothing (Moving average, 3 segments)	7	89.0	0.3266	88.9	0.3270
MSC	3	85.3	0.3774	81.3	0.4734

R^2^: the coefficient of determination; LVs: latent variables; RMSEC: root mean square error of calibration; RMSEP: root mean square error of prediction; MSC: multiplicative scatter correction; EMSC: extended multiplicative scatter correction. The results of the best functioning models are displayed in Appendix A.

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
