# Peer review of "Reliable Discrimination of Green Coffee Beans Species: A Comparison of UV-Vis-Based Determination of Caffeine and Chlorogenic Acid with Non-Targeted Near-Infrared Spectroscopy"

_foods, 2020, doi:10.3390/foods9060788_

Round 1

Reviewer 1 Report

Reviewer comments

Introduction

The introduction is poorly written, lacks information and does not highlight clearly the significance/novelty of the research. The introduction should be re-written and re-structured. More specific details can be found below;

  1. Line 32-38: It would be beneficial for readers if the authors could include more facts on the global coffee trade i.e. increasing demand for coffee bean production, consumption rates, economic significance, most popular coffee beans (e.g. Arabica, Robusta) etc. Then explain how these two types differ in terms of characteristics, composition, cost, which species is deemed higher quality and why (taste, aroma etc.).
  2. From a food fraud perspective the problem is not clear from the first sentence. What is coffee commonly adulterated with? Why is it important to assess the authenticity of coffee (or any food product)? In this research, is Robusta considered a possible adulterant of Arabica, or vice versa? Does the reader assume that Arabica is authentic and Robusta is the adulterant, based on the cost?
  3. Line 35-38: ‘the mean liking scores in a consumer test…many consumers prefer Arabica to Robusta.’ Is the consumer test based on taste, aroma, appearance etc.? Readers shouldn’t have to look through the references to find this information.
  4. Line 54-56: more information on the two compounds; caffeine and chlorogenic acid. What sort of compounds are they? What effects do they have on the body when consumed? Any health benefits/concerns? Which one has more effect on the quality of coffee? Then mention the differences in compound content between Arabica and Robusta.
  5. Line 65-66: the author needs to be careful when making such claims. Reference 14 refers to a poster which has now been published as an article [1]. The reference list should include the article (see below).
  6. UV-Vis and NIR are both well-known techniques in this research area. NIR is commonly used to identify food fraud based on geographical origin, species and variety. The author needs to highlight the novelty/originality of their research clearly. Perhaps the main focus of the manuscript should be on comparing the two spectroscopic techniques, as stated in the title.

Materials and Methods

  1. Might be useful to include equations in either the main manuscript or supplementary information (e.g. Beer-Lambert, % dw, RMSEC or RMSEP).

Results

  1. The main focus of the manuscript is UV-Vis and NIR, but the raw spectra has not been included in either the main text or supplementary information. The spectra of standards should also be included.

Spelling and grammar

  1. Line 41: ‘between varieties within species’ change to ‘between species and variety’ throughout manuscript.
  2. Line 47: sentence does not read well. Please reword e.g. Analytical methods such as, chromatographic and spectroscopic techniques are commonly applied successfully to discriminate between coffee bean species.
  3. Line 67-68: ‘to avoid any fraud in trade with green coffee beans’. Should be changed to ‘to help prevent fraud within the global coffee bean trade.’
  4. Line 71: ‘considerably’ should be changed to ‘considerable.’
  5. Line 387: consider changing word ‘superior’ to ‘comparable.’
  6. Line 396: ‘fits’ should be changed to ‘fit.’
  7. Line 404: ‘the scatter effect’ should be changed to ‘scattering effect’ or just ‘scattering.’
  8. Line 406: ‘Various varieties within species’ should be changed to ‘species variety can.’
  9. Line 407-408: ‘various varieties within the examined species’ should be changed to ‘different varieties within the species analysed’ or ‘different species and variety may lead to scattering problems.’
  10. Line 409: word ‘on’ change to ‘during.’
  11. Line 433: ‘consist of various varieties within one species’ should be changed to ‘display variations amongst species and variety.’
  12. Line 434-435: ‘varieties at the same location’ should be changed to ‘batches at the same time.’

Other comments

  1. More detail on future work should be mentioned in the conclusion i.e. adulteration study using this approach or studying the species, variety and geographical origin to determine authenticity, using larger data sets and authentic/adulterant samples from different locations.
  2. Ensure that the supplementary information numbering is in chronological order throughout the main text.

References

  1. Navarra, G., et al., Simultaneous Determination of Caffeine and Chlorogenic Acids in Green Coffee by UV/Vis Spectroscopy. Journal of Chemistry. 2017: p. 6435086.

Comments can also be found in the attached document.

Author Response

Introduction

The introduction is poorly written, lacks information and does not highlight clearly the significance/novelty of the research. The introduction should be re-written and re-structured. More specific details can be found below;

  1. Line 32-38: It would be beneficial for readers if the authors could include more facts on the global coffee trade i.e. increasing demand for coffee bean production, consumption rates, economic significance, most popular coffee beans (e.g. Arabica, Robusta) etc. Then explain how these two types differ in terms of characteristics, composition, cost, which species is deemed higher quality and why (taste, aroma etc.).

Reply: We agree with the reviewer and have added more information regarding coffee production and how Arabica and Robusta differ (line 34-48).

  1. From a food fraud perspective the problem is not clear from the first sentence. What is coffee commonly adulterated with? Why is it important to assess the authenticity of coffee (or any food product)? In this research, is Robusta considered a possible adulterant of Arabica, or vice versa? Does the reader assume that Arabica is authentic and Robusta is the adulterant, based on the cost?

Reply: We follow the reviewer´s suggestion. We have explained in the first paragraph that there is a price gap between Arabica and Robusta which may lead to fraud. (line 34-36). Robusta is considered as an adulterant to Arabica (line 48), but such fraud also includes the addition of low-cost materials and coffee beans from different geographical regions (line 39-41).

  1. Line 35-38: ‘the mean liking scores in a consumer test…many consumers prefer Arabica to Robusta.’ Is the consumer test based on taste, aroma, appearance etc.? Readers shouldn’t have to look through the references to find this information.

Reply: We thank the reviewer for this helpful suggestion. We have added the information that the consumer test was based on aroma, flavor, and mouthfeel (line 44).

  1. Line 54-56: more information on the two compounds; caffeine and chlorogenic acid. What sort of compounds are they? What effects do they have on the body when consumed? Any health benefits/concerns? Which one has more effect on the quality of coffee? Then mention the differences in compound content between Arabica and Robusta.

Reply: We agree with the reviewer´s opinion and have added more information regarding caffeine and chlorogenic acid (line 57-64).

  1. Line 65-66: the author needs to be careful when making such claims. Reference 14 refers to a poster which has now been published as an article [1]. The reference list should include the article (see below).

Reply: Thank you for pointing that out. We have corrected the claim to ‘no previous study has so far discriminated among green coffee bean species used UV-Vis spectroscopy based only to these two compounds’ (line 85-86). The reference has been added to the article (reference no 21).

  1. UV-Vis and NIR are both well-known techniques in this research area. NIR is commonly used to identify food fraud based on geographical origin, species and variety. The author needs to highlight the novelty/originality of their research clearly. Perhaps the main focus of the manuscript should be on comparing the two spectroscopic techniques, as stated in the title.

Reply: We agree with the opinion of the reviewer and have added the following sentence …’in comparison with the UV-Vis based determination of caffeine and chlorogenic acid’ as suggested (line 94-95).

Materials and Methods

  1. Might be useful to include equations in either the main manuscript or supplementary information (e.g. Beer-Lambert, % dw, RMSEC or RMSEP).

Reply: Thanks for the advice. We have added equations as supplementary materials in Table S1.

Results

  1. The main focus of the manuscript is UV-Vis and NIR, but the raw spectra has not been included in either the main text or supplementary information. The spectra of standards should also be included.

Reply: Many thanks also for this advice. The raw (mean) spectra for Arabica and Robusta are presented (Figure 3a).

Spelling and grammar

  1. Line 41: ‘between varieties within species’ change to ‘between species and variety’ throughout manuscript.
  2. Line 47: sentence does not read well. Please reword e.g. Analytical methods such as, chromatographic and spectroscopic techniques are commonly applied successfully to discriminate between coffee bean species.
  3. Line 67-68: ‘to avoid any fraud in trade with green coffee beans’. Should be changed to ‘to help prevent fraud within the global coffee bean trade.’
  4. Line 71: ‘considerably’ should be changed to ‘considerable.’
  5. Line 387: consider changing word ‘superior’ to ‘comparable.’
  6. Line 396: ‘fits’ should be changed to ‘fit.’
  7. Line 404: ‘the scatter effect’ should be changed to ‘scattering effect’ or just ‘scattering.’
  8. Line 406: ‘Various varieties within species’ should be changed to ‘species variety can.’
  9. Line 407-408: ‘various varieties within the examined species’ should be changed to ‘different varieties within the species analysed’ or ‘different species and variety may lead to scattering problems.’
  10. Line 409: word ‘on’ change to ‘during.’
  11. Line 433: ‘consist of various varieties within one species’ should be changed to ‘display variations amongst species and variety.’
  12. Line 434-435: ‘varieties at the same location’ should be changed to ‘batches at the same time.’

Reply: All spelling and grammar remarks have been changed. Thank you very much for the suggestions to improve our manuscript.

Other comments

  1. More detail on future work should be mentioned in the conclusion i.e. adulteration study using this approach or studying the species, variety and geographical origin to determine authenticity, using larger data sets and authentic/adulterant samples from different locations.

Reply: Thanks for the advice and suggestions. We have added more information as suggested (line 468-474).

  1. Ensure that the supplementary information numbering is in chronological order throughout the main text.

Reply: We have check the supplementary materials are in chronological order.

Reviewer 2 Report

The authors reported an interesting and valuable work in the manuscript titled “Reliable Discrimination of Green Coffee Bean 2 Species: A comparison of UV-Vis based 3 determination of Caffeine and Chlorogenic Acid with 4 Non-targeted Near-infrared Spectroscopy”. Generally the manuscript was well written and organized. However there are several issues need to deal with before the manuscript could be accepted for publications. Generally I suggest a minor revision of the manuscript. The detailed comments are as follow:

  1. The author should describe the aims of this study more clearly in the end of the part of Introduction.
  2. In the part of Materials and methods, the author should add a figure to plot the study area of this research and then make a brief introduction of the study area.
  3. Figure 4(a) is too blur to read, please replaced it with a figure having higher resolution
  4. Could the authors explained the samples located in the left bottom corner of the figure 5
  5. Could the author try to use the Support Vector Machines to make prediction and then compare the prediction accuracy with current methods?

Author Response

Comments and Suggestions for Authors

The authors reported an interesting and valuable work in the manuscript titled “Reliable Discrimination of Green Coffee Bean 2 Species: A comparison of UV-Vis based 3 determination of Caffeine and Chlorogenic Acid with 4 Non-targeted Near-infrared Spectroscopy”. Generally the manuscript was well written and organized. However there are several issues need to deal with before the manuscript could be accepted for publications. Generally I suggest a minor revision of the manuscript. The detailed comments are as follow:

1. The author should describe the aims of this study more clearly in the end of the part of Introduction.

Reply: We agree with the reviewer and have added the sentences at the end of the paragraph (line 95-99).

2. In the part of Materials and methods, the author should add a figure to plot the study area of this research and then make a brief introduction of the study area.

Reply: We have added a map of the sampling area as supplementary materials (Figure S1). We would like to focus on species discrimination in the manuscript body. This is because we have put a focus on the origin identification by using the same samples, but a different identification method in another manuscript.

3. Figure 4(a) is too blur to read, please replaced it with a figure having higher resolution

Reply: Figures 4a and 4b have been replaced with higher resolution image.

4. Could the authors explained the samples located in the left bottom corner of the figure 5

Reply: We check more details to the sample located in the left bottom corner of Figure 5. This sample (black line) shows a different spectral pattern as compared to other Robusta samples. This particular sample did not dry well on the site and may have been infested by fungi. Thus, these circumstances may explain the odd pattern for the sample.

Please also take a look at the included Figure in the Cover letter on page 4.

5. Could the author try to use the Support Vector Machines to make prediction and then compare the prediction accuracy with current methods?

Reply: We prefer to stick with PLS_DA and LDA, especially because these methods perform satisfactorily and their results can be interpreted (i.e. spectral contributions in case of PLS-DA.

Please also take a look at the included Figure in the Cover letter on page 5.

Reviewer 3 Report

The manuscript by Adnan Adnan and colleagues provides comprehensive assessments of two spectroscopic techniques targeting a reliable discrimination of green coffee bean species.

The work is important in that it probes a highly potential and inexpensive modes of coffee been characterization.

Rational experimental design, good technical level and straightforward logic are supported by all required technical details, adequate presentation, and good statistical level.

Conclusions are fully supported. Sample size is not too big but was sufficient to build validated statistical models.  

Assessment of the degree and character of patterns of spectral variations is standard. It corresponds to the selected level of consideration and conclusions.

Author Response

Comments and Suggestions for Authors

The manuscript by Adnan Adnan and colleagues provides comprehensive assessments of two spectroscopic techniques targeting a reliable discrimination of green coffee bean species.

The work is important in that it probes a highly potential and inexpensive modes of coffee been characterization.

Rational experimental design, good technical level and straightforward logic are supported by all required technical details, adequate presentation, and good statistical level.

Conclusions are fully supported. Sample size is not too big but was sufficient to build validated statistical models.

Assessment of the degree and character of patterns of spectral variations is standard. It corresponds to the selected level of consideration and conclusions.

Reply: Thanks to the reviewer for the excellent feedback.

Round 2

Reviewer 1 Report

Thank you.

The authors have addressed all of my comments and made all of the recommended changes.

I now believe the manuscript to be more relevant, interesting, informative and original.

Therefore, I am happy to accept the manuscript for publication in Foods without further changes.

Best regards.

Author Response

Reply: Thanks to the reviewer for the excellent feedback.